## [Peer Review File · Development (Cambridge, England)]

ShineGal4 drivers for tissue and cell-type specific optogenetics in *Drosophila*

Victor Girard, Sebastian Sorge, Joachim Kurth, Cyrille Alexandre and Alex P. Gould
DOI: 10.1242/dev.204981

Editor: Thomas Lecuit

Review timeline

Original submission:	27 May 2025
Editorial decision:	1 July 2025
First revision received:	12 December 2025
Accepted:	8 January 2026

Original submission

First decision letter

MS ID#: dev.204981

MS TITLE: ShineGal4 drivers for tissue and cell-type specific optogenetics in *Drosophila*

AUTHORS: Victor Girard, Sebastian Sorge, Joachim Kurth, Cyrille Alexandre and Alex P. Gould

Dear Alex,

I have now received all the referees reports on the above manuscript, and have reached a decision. The referees' comments are appended below, or you can access them online: please go to .

The overall evaluation is positive and we would like to publish a revised manuscript in *Development*, provided that the referees' comments can be satisfactorily addressed. Please attend to all of the reviewers' comments in your revised manuscript and detail them in your point-by-point response. If you do not agree with any of their criticisms or suggestions explain clearly why this is so. If it would be helpful, you are welcome to contact us to discuss your revision in greater detail. Please send us a point-by-point response indicating your plans for addressing the referees' comments, and we will look over this and provide further guidance.

Reviewer 1

Advance summary and potential significance to field

This paper makes an invaluable contribution to the field of *Drosophila* genetics that allows further refinement of space and time controlled genetic manipulations in multiple essential tissues. These new tool kit and further derivations of it would open up many new possibilities to developmental and adult tissue studies in the field. This is especially meaningful for the study of essential gene function in very discrete cell populations in organs with heterogeneous cell populations, such as the adult intestine. Thanks for generating such beautiful tools for the community!!!!

Comments for the author

- 1-The paper will be greatly enriched by the inclusion of quantifications. Including comparative analysis of transgene expression in the different tissues at 4hs versus 24hs after light-induced transgene activation.
- 2- Also, efficiency of the system should be presented in a quantitative manner. For instance, how much of a tissue shows transgene activation and what proportion of a population of animals/tissues exposed to light show transgene activity at a given time point.
- 3- In addition to CD, it would be great if transgene of known cellular function could be expressed in the flip out system as a positive control. For example a transgene that would induce cell growth or shrinkage could be a nice example. Or the activation of a signalling event that could be verified by antibody/reporter staining.

Reviewer 2

Advance summary and potential significance to field

This is a short and straightforward resource paper. Applying a previously reported method of genome engineering to a previously reported optogenetic variant of Gal4, a series of Gal4 lines were repurposed to confer light-dependent control onto the activity of Gal4. Although novelty is limited, the paper will be of interest to the fly community and may deserve to be published in Development once the following question has been addressed : what is the transcriptional activity of these new shine-gal4 lines relative to their Gal4 counterparts ? Are these shine-gal4 as strong as the original Gal4 lines ?

Since GFP and other fluorescent protein are relatively stable, I would suggest to perform this analysis at the mRNA level using a UAS-X transgene producing a relatively unstable RNA. It would also be nice to perform this quantitative analysis at different time points after light-induction to inform the dynamics of light-dependent increase of shine-gal4 activity over time (unless it is already known that shine-Gal4 become fully active instantaneously).

Reviewer 3

Advance summary and potential significance to field

This paper from Girard et al merges the recent developments in light-controlled genetic activation with the extensive collection of binary genetic tools (GAL4/UAS) in Drosophila. This results in a new, publicly accessible collection of light-inducible drivers for various cell populations, with a focus on the central nervous system. Using a previously published in situ gene replacement method (HACK), the authors first convert a series of well-characterized GAL4 driver lines for different cell population into Shine-GAL4 driver lines. In these drivers, Magnet photoswitches assemble a functional GAL4 form upon light exposure. The authors further validate the inducibility and minimal leakiness of these Shine-GAL4 lines by assessing the expression of a fluorescent reporter with and without light exposure. They finally propose a refined, clonal version of the Shine-GAL4 principle by coupling it with the Flp-Out mosaic technique. This combination enables the separation of clonal induction and functional activation.

The ability to precisely control when and where genetic manipulations occur is crucial for establishing causality and a skill that biologists continually need to improve. Shine-GAL4 has been shown (Di Pietro et al, 2021) to provide faster and more spatially precise manipulations than conventional methods (using temperature or hormonal control) thanks to the on/off control of a directed light source. By providing a collection of novel Shine drivers, this study is of clear interest to the Drosophila community, and in particular to neurobiologists. The methodology and rationale, although not new, are well-executed, and the clonal approach brings a refinement to the system.

Comments for the author

Major comment

The authors justify the generation of Shine-GAL4 lines in general, and especially in the nervous system, by citing improved temporal (< 7-10h of the TARGET system) and spatial (defined by light exposure perimeter) sensitivities. However, for the drivers targeting cell populations of the central

nervous system (the core collection of this study), only the result after 24 h exposition is shown, and at the scale of the whole cell population (Figure 2). I understand that the purpose of this study is not to recapitulate the findings from the original paper supporting the general advantages of the Shine-GAL4 system (Di Pietro et al, 2021). However, I think it is important to provide evidence supporting the use of this specific collection for other scientists. This information would help them know the temporal and spatial sensitivity they can expect from these newly generated drivers, helping in their choice between systems and serving as a reference for their own use of the lines. As such, I would suggest that the authors show the results of 4 h (or less than 7 h) of light exposure of the induction for these 7 lines. I would also propose to illustrate the spatial sensitivity by testing activation at the subpopulation level (for example, half of the central nervous system) for a subset of these lines.

Minor comments

1. It would be useful if the authors could explain why/when separating clonal induction and system activation would be advantageous. While they mention that this approach « may be particularly useful for analysing transgenes with severe phenotypes », it is unclear why growing silent (for the Shine-GAL4 system or another system at the same later time point. One possible advantage I can think of could be the ability to simultaneously activate an entire lineage whose cells have inherited the original flip-out, whereas conventional mosaic induction at a later time point would result in sparse lineage coverage -so this approach would be useful for proliferating cells. Could the authors elaborate on this ?

2. The GFP signal varies in non-induced (dark/no light conditions). For example, PromE-ShineGAL4 is stronger than Mex1-ShineGAL4 (Figure 1), and alm-ShineGAL4 is much stronger than moody-ShineGAL4 (Figure 2). Could the authors comment on whether the GFP signal detected in the dark is genuine (and then how to take it into account), or if it is only some auto-fluorescence ? Is it because some were tested before the removal of the 3xP3-Cherry cassette? The authors mention this cassette gives some leakiness for the nMag part, however it is not shown.

First revision

Author response to reviewers' comments

Reviewer #1:

This paper makes an invaluable contribution to the field of *Drosophila* genetics that allows further refinement of space and time controlled genetic manipulations in multiple essential tissues. These new tool kit and further derivations of it would open up many new possibilities to developmental and adult tissue studies in the field. This is especially meaningful for the study of essential gene function in very discrete cell populations in organs with heterogeneous cell populations, such as the adult intestine. Thanks for generating such beautiful tools for the community!!!!

We thank the Reviewer for their kind words.

1-The paper will be greatly enriched by the inclusion of quantifications. Including comparative analysis of transgene expression in the different tissues at 4hs versus 24hs after light-induced transgene activation.

We now provide quantifications of GFP fluorescence at 4 time points (0, 3, 6 and 24 hours of light exposure) for 9 of the 14 ShineGal4 drivers. *Lpp-ShineGal4* (fat body) and *PromE-ShineGal4* (oenocytes) are shown in **Figure S1 (C-F)** and the seven CNS drivers are shown in **Figure S2**. After 3 hr of driver photoactivation, GFP fluorescence is detected with most ShineGal4 drivers indicating rapid optogenetic induction. GFP fluorescence increases with the duration of photoactivation and, by 24h, the levels induced by about half of the ShineGal4 drivers are comparable to those of the original GAL4 drivers (new **Figure S3**).

2- Also, efficiency of the system should be presented in a quantitative manner. For instance, how much of a tissue shows transgene activation and what proportion of a population of animals/tissues exposed to light show transgene activity at a given time point.

The reviewer raises an important point. Our experimental protocol uses only a very thin layer of food so that all larvae are exposed to the light source and variable expression between larvae has not been an issue. Nevertheless, cell-to-cell variability in expression levels within some ShineGAL4 driver spatial domains has been reported at early time points after light induction. Therefore, in the revised manuscript we provide new images in **Figures S1C** and **S1E** of whole larvae for the fat body driver (*Lpp-ShineGAL4*) and broader views for the oenocyte driver (*PromE-ShineGAL4*). In both cases, we see that GFP is strongly induced throughout the spatial domain of the optogenetic driver, akin to that seen with the original GAL4 counterparts. To provide an honest realistic guide for future users, we also compared side-by-side the GFP fluorescence patterns of nine ShineGal4 drivers (after 24 hr of light induction) and their original Gal4 counterparts (new **Figure S3**). In each case, the spatial patterns appear similar but, for about half of the drivers, the expression is weaker in the ShineGAL4 version, at least after 24 hr of light induction. Hence, the ShineGAL4 drivers for *PromE*, *Lpp*, *nab*, *Cyp4g15*, and *moody* give comparable expression to their GAL4 counterparts whereas it is weaker for *repo* and *alm*, and for *elav* it is only strong within a subset of its original Gal4 domain. We suspect that the differences in expression levels are due either to accumulation of GFP expressed from the original GAL4 drivers over a longer period of time than 24 hr and/or to cell-type differences in expression levels of the ubi-AD-Mag split driver. Either way, we get enough expression to be detected by GFP fluorescence and more strongly by anti-GFP antibody (**Figure 2** and new **Figure S2**).

3- In addition to CD, it would be great if transgene of known cellular function could be expressed in the flip out system as a positive control. For example a transgene that would induce cell growth or shrinkage could be a nice example. Or the activation of a signalling event that could be verified by antibody/reporter staining.

We thank the reviewer for their suggestion. As a proof-of-principle, we manipulated the tuberous sclerosis complex genes *TSC1* and *TSC2*, well characterized negative regulators of cell size in the fat body (for example, **Figure 3c** in Scott *et al.* 2004 PMID: 15296714). FLP-out silent clones with the potential to co-express *UAS-TSC1/2* were induced in the fat body and nuclear volume was measured (new **Figure 3D-E**). In “silent” clones (generated in the dark), nuclear volumes remained unaffected and comparable to those of surrounding control cells, whereas the size of nuclei in clones exposed to light for 24 or 48 hr decreased in a light-dose dependent manner. This experiment indicates that ShineGAL4 FLP-out technology can be used to separate in time the events of clone induction and GAL4 activation, so that the acute effects of severe phenotypes can be studied in viable clones. Importantly, this method allows direct comparisons of “silent” and “photoactivated” clones with identical genetic backgrounds.

Reviewer #2:

This is a short and straightforward resource paper. Applying a previously reported method of genome engineering to a previously reported optogenetic variant of Gal4, a series of Gal4 lines were repurposed to confer light-dependent control onto the activity of Gal4. Although novelty is limited, the paper will be of interest to the fly community and may deserve to be published in Development once the following question has been addressed:

what is the transcriptional activity of these new shine-gal4 lines relative to their Gal4 counterparts ? Are these shine-gal4 as strong as the original Gal4 lines ? Since GFP and other fluorescent protein are relatively stable, I would suggest to perform this analysis at the mRNA level using a UAS-X transgene producing a relatively unstable RNA. It would also be nice to perform this quantitative analysis at different time points after light-induction to inform the dynamics of light- dependent increase of shine-gal4 activity over time (unless it is already known that shine-Gal4 become fully active instantaneously).

We thank the Reviewer for their comments. In line with these, we have compared side-by-side the new ShineGal4 lines to their Gal4 counterparts using the same *UAS-GFP* reporter (also see our

response to Reviewer #1). To provide an honest realistic guide for future users, we compared side-by-side the GFP fluorescence patterns of nine ShineGal4 drivers (after 24 hr of light induction) and their original Gal4 counterparts (new **Figure S3**). In each case, the spatial patterns appear similar but for about half of the drivers, the expression is weaker in the ShineGAL4 versions, at least after 24 hr of light induction. Hence, the ShineGAL4 drivers for *PromE*, *Lpp*, *nab*, *Cyp4g15* and *moody* give comparable expression to their GAL4 counterparts whereas it is weaker for *repo* and *alrm*, and for *elav* it is only strong within a subset of its Gal4 domain. We suspect that the differences in expression levels are due either to accumulation of GFP expressed from the original GAL4 drivers over a very long period of time (greater than 24 hr) and/or to cell-type differences in expression levels of the ubi-AD-mag split driver. Either way, expression is sufficient to be detectable by GFP fluorescence and more strongly by anti-GFP antibody (**Figure 2** and new **Figure S2**). We also provide evidence that ShineGAL4 expression levels are sufficient to generate clonal phenotypes using the new ShineGAL4 FLP-out construct (new **Figure 3D-E**). This adds to other evidence of ShineGAL4 gain and loss-of-function phenotypes at animal, organ, and cellular levels (di Pietro *et al.* 2021 PMID:[34879263](https://pubmed.ncbi.nlm.nih.gov/34879263/)).

We have addressed the dynamics of the light-dependent increase of ShineGal4 activity by quantifying GFP fluorescence at four time points (0, 3, 6 and 24 hours of light exposure) for nine of the fourteen ShineGal4 drivers. *Lpp-ShineGal4* (fat body) and *PromE-ShineGal4* (oenocytes) are shown in new **Figure S1 (C-F)** and the seven CNS drivers are shown in new **Figure S2**. After 3 hr of light exposure, GFP fluorescence is detected with most ShineGal4 drivers indicating rapid optogenetic induction. GFP fluorescence increases with the duration of photoactivation and, by 24h, the levels of GFP fluorescence induced by about half of ShineGal4 drivers are comparable to their original GAL4 counterparts (new **Figure S3**). The Reviewer suggested to use an unstable RNA system but we have not been able to set this up in the time available. We nevertheless refer the Reviewer to the unstable protein reporter (UAS- nls:Scarlet:PEST) that has already been used to determine the switch-off kinetics of Shine-GAL4. In this reporter context, expression was maximal after 7 +/-1 hr with the *ubi-ShineGAL4* system (di Pietro *et al.* 2021 PMID:[34879263](https://pubmed.ncbi.nlm.nih.gov/34879263/)).

Reviewer #3:

This paper from Girard et al merges the recent developments in light-controlled genetic activation with the extensive collection of binary genetic tools (GAL4/UAS) in *Drosophila*. This results in a new, publicly accessible collection of light-inducible drivers for various cell populations, with a focus on the central nervous system. Using a previously published in situ gene replacement method (HACK), the authors first convert a series of well-characterized GAL4 driver lines for different cell population into Shine-GAL4 driver lines. In these drivers, Magnet photoswitches assemble a functional GAL4 form upon light exposure. The authors further validate the inducibility and minimal leakiness of these Shine-GAL4 lines by assessing the expression of a fluorescent reporter with and without light exposure. They finally propose a refined, clonal version of the Shine-GAL4 principle by coupling it with the FLP-Out mosaic technique. This combination enables the separation of clonal induction and functional activation.

The ability to precisely control when and where genetic manipulations occur is crucial for establishing causality and a skill that biologists continually need to improve. Shine-GAL4 has been shown (Di Pietro et al, 2021) to provide faster and more spatially precise manipulations than conventional methods (using temperature or hormonal control) thanks to the on/off control of a directed light source. By providing a collection of novel Shine drivers, this study is of clear interest to the *Drosophila* community, and in particular to neurobiologists. The methodology and rationale, although not new, are well-executed, and the clonal approach brings a refinement to the system.

We thank the Reviewer for their comments.

Major comment

The authors justify the generation of Shine-GAL4 lines in general, and especially in the nervous system, by citing improved temporal (< 7-10h of the TARGET system) and spatial (defined by light exposure perimeter) sensitivities. However, for the drivers targeting cell populations of

the central nervous system (the core collection of this study), only the result after 24 h exposition is shown, and at the scale of the whole cell population (Figure 2). I understand that the purpose of this study is not to recapitulate the findings from the original paper supporting the general advantages of the Shine-GAL4 system (Di Pietro et al, 2021). However, I think it is important to provide evidence supporting the use of this specific collection for other scientists. This information would help them know the temporal and spatial sensitivity they can expect from these newly generated drivers, helping in their choice between systems and serving as a reference for their own use of the lines.

As such, I would suggest that the authors show the results of 4 h (or less than 7 h) of light exposure of the induction for these 7 lines. I would also propose to illustrate the spatial sensitivity by testing activation at the subpopulation level (for example, half of the central nervous system) for a subset of these lines.

To address directly the Reviewer's comments, we have now quantified GFP fluorescence at 4 time points (0, 3, 6 and 24 hours of light exposure) for nine of the fourteen ShineGal4 drivers. *Lpp-ShineGal4* (fat body) and *PromE-ShineGal4* (oenocytes) are shown in Figure S1 (C-F) and the seven CNS drivers are shown in Figure S2. After 3 hr of light exposure, GFP fluorescence is detected with most ShineGal4 drivers and there is much stronger expression after 24 hours. Regarding the spatial sensitivity/specificity of the Shine-Gal4 system, this has already been elegantly demonstrated by localized photoactivation in a previous study (di Pietro *et al.* 2021 PMID: [34879263](https://pubmed.ncbi.nlm.nih.gov/34879263/)). These experiments used a sophisticated live imaging system with a two-photon microscope that we are not currently set up for. In any case, we envisage that for most applications of our new shineGAL4 collection, spatial specificity will be provided by the driver itself, with uniform photoactivation being primarily used to regulate timing.

Minor comments

1. It would be useful if the authors could explain why/when separating clonal induction and system activation would be advantageous. While they mention that this approach « may be particularly useful for analysing transgenes with severe phenotypes », it is unclear why growing silent (for the Shine-controlled construct) clones and activating them later would differ from simply activating the Shine-GAL4 system or another system at the same later time point. One possible advantage I can think of could be the ability to simultaneously activate an entire lineage whose cells have inherited the original flip-out, whereas conventional mosaic induction at a later time point would result in sparse lineage coverage -so this approach would be useful for proliferating cells. Could the authors elaborate on this?

We thank the Reviewer for raising this important point and we have now elaborated on it in the Results and Discussion of the revised manuscript (also see our response to Reviewer #1). We have also conducted a new functional experiment to illustrate how the ShineGAL4 FLP-out method can be used to regulate cell size in the fat body in a light-dose dependent manner (new Figure 3D-E). This experiment indicates that ShineGAL4 FLP-out can be used to separate in time the events of clone induction and GAL4 activation, so that the acute effects of severe phenotypes can be studied in viable clones. As the Reviewer wisely comments, the method would also be useful for proliferating cells and we have also added this point to the Results and Discussion.

2. The GFP signal varies in non-induced (dark/no light conditions). For example, *PromE-ShineGAL4* is stronger than *Mex1-ShineGAL4* (Figure 1), and *alarm-ShineGAL4* is much stronger than *moody-ShineGAL4* (Figure 2). Could the authors comment on whether the GFP signal detected in the dark is genuine (and then how to take it into account), or if it is only some auto-fluorescence? Is it because some were tested before the removal of the 3xP3-Cherry cassette? The authors mention this cassette gives some leakiness for the nMag part, however it is not shown.

We acknowledge the confusing nature of the signal in the anti-GFP immunostainings for the “dark” controls in Figure 1 and Figure 2 of the original manuscript. This is due to background staining/autofluorescence not to real mGFP expression. In line with this, we provide new experimental data showing that there is little or no “dark” signal observed with native GFP fluorescence rather than antibody staining (revised Figure S1 and new Figure S2). Hence *PromE-ShineGal4* at 0 hr (i.e in the dark) shows almost no fluorescence on the GFP channel (revised

Figure S1E-F). For *alm-Gal4*, the background in the original “dark” immunostaining (**Figure 2G**) was higher than for other CNS drivers as the laser power used was higher, in order to match that needed to capture the low level signal from mGFP with this rather weak driver (see revised **Figure S3E**).

We have now clarified this issue in the revised manuscript, in the legends of Figures 1 and 2.

Regarding the 3xP3-Cherry cassette, we found that it gave leakiness in one of the ShineGAL4 drivers (*PromE-ShineGAL4*). The other important reason that the selection cassette was removed from most ShineGAL4 drivers was to prevent unwanted signal in the red channel as the 3xP3-Cherry cassette is expressed in the larval CNS, mainly in the cortex glia. However, for three of the now fourteen drivers (*nSyb-ShineGAL4*, *uro-ShineGal4* and *Mex1-ShineGal4*) we retained the 3xP3-Cherry cassette as we found that its excision impaired driver expression. This is now explained in the revised manuscript and we have also clarified in the genotypes detailed in the revised figure legends when the 3xP3-Cherry cassette was present and when it was excised.

Second decision letter

MS ID#: dev.204981R1

MS TITLE: ShineGal4 drivers for tissue and cell-type specific optogenetics in Drosophila

AUTHORS: Victor Girard, Sebastian Sorge, Joachim Kurth, Cyrille Alexandre and Alex P. Gould

Dear Alex,

I am happy to tell you that your manuscript has been accepted for publication in Development, pending our standard publication integrity checks.

Reviewer 1

Advance summary and potential significance to field

The authors have satisfactorily addressed my comments on the original submission. These lines will be a very valuable tool for the community. Congratulations!

Reviewer 2

Advance summary and potential significance to field

the authors have adequately addressed the comments made by the reviewers in the revised version of the manuscript

Reviewer 3

Advance summary and potential significance to field

I first thank the authors for having taken the time to answer my comments. I am satisfied with their replies, either through experiments (the functional Tsc1/2 experiment is especially elegant and well chosen to illustrate the use of silent clones) or writing.

I understand the technical constraint for performing regional activation. Ultimately future users will have to determine the specific advantage of a chosen ShineGAL4 driver for their own experimental set-up, and the guidelines provided by the paper form a sufficient basis.

Minor comments

- I would suggest underlining in the main text the difference between Figure 2 and Figure S2 regarding the use of an anti-GFP staining in Figure 2. It would prevent interrogations about the background (which are answered in the Replies to reviewers but not so much in the manuscript itself).

- I am also wondering whether Figures 2 and S2 should be swapped. I understand that the overall signal is clearer in Figure 2, yet S2 brings temporality and quantifications (and more direct comparisons with Figure S3 where no anti-GFP was used if I understood correctly).